# Rationale and Purpose: The FLUTE Study to Evaluate Fluorography Mass Screening for Tuberculosis and Other Diseases, as Conducted in Eastern Europe and Central Asia Countries

**DOI:** 10.3390/ijerph19148706

**Published:** 2022-07-17

**Authors:** Vitaly Smelov, Olga Trusova, Sylvaine Barbier, Richard Muwonge, Viatcheslav Grankov, Valiantsin Rusovich, Armando Baena, Mary Lyn Gaffield, Marilys Anne Corbex, Masoud Dara

**Affiliations:** 1World Health Organization (WHO) Regional Office for Europe, 2100 Copenhagen, Denmark; smelovv@who.int (V.S.); corbexm@who.int (M.A.C.); 2Early Detection, Prevention & Infections Branch, International Agency for Research on Cancer (IARC) WHO, 69372 Lyon, France; sylvaine.barbier@gmail.com (S.B.); muwonger@iarc.fr (R.M.); baenaa@iarc.fr (A.B.); 3World Health Organization (WHO) Country Office, 220007 Minsk, Belarus; grankovv@who.int (V.G.); rusovichv@who.int (V.R.); masoud_dara@yahoo.com (M.D.); 4World Health Organization (WHO) Headquarters, 1211 Geneva, Switzerland; gaffieldm@who.int

**Keywords:** lung cancer, pulmonary tuberculosis, fluorography, chest X-ray, dispanserization, Belarus, screening

## Abstract

In Belarus and several EECA countries, periodic population-based chest X-ray “fluorography programme” use as a mass screening tool for the diagnosis of tuberculosis (TB) has been used for decades. This mass screening has also often been justified for the early detection of lung cancer (LC), although no mortality benefits were demonstrated by screening with chest X-ray in international randomized trials. In Belarus, fluorography testing is mandatory every one to three years for all adults depending on age and the so-called “risk groups”. The World Bank and WHO estimate that Belarus spends USD11 million annually on mass fluorography screening and advocate for more targeted screening approaches to increase diagnostic yield for TB and not to use it for screening for LC. The study is a retrospective review of medical records to assess the yield of fluorography to detect true cases of LC and/or TB in asymptomatic patients in two rural and two urban districts in Belarus for 2015–2017 with positive screening results for presumed of TB or LC. The study provided the rationale to implement the improved policy and practices regarding the role of fluorography in the early detection of LC and TB in Belarus and elsewhere.

## 1. Introduction

In the Soviet Union (1922–1991), of which Belarus was a part, tuberculosis (TB) was considered a socially significant disease and widespread among the population, especially during and after the revolutions and wars. For instance, the impact of World War II exacerbated the situation of pulmonary TB, bringing the incidence in 1950 to 290 people per 100,000 [1] and the death rate in 1941 to 80/100,000 [2], prompting the active introduction of measures for earlier and more accurate detection than clinical examination. Thus, the mass use of chest X-ray examination became a common practice.

At that time, the introduction of mass chest X-ray screening for the early detection of TB, together with increasing access to anti-TB treatment, gradual improvements in well-being, and housing conditions, contributed to a reduction in TB incidence and mortality. The number of registered cases decreased from 290/100,000 in 1950 to 37/100,000 in 2017 and the mortality rate lowered from 80/100,000 in 1941 to 1.4/100,000 in 2017 [1,2]. However, due to multiple factors including an inadequate treatment regimen, prolonged hospital treatment with insufficient infection control measures, stock-outs of the anti-TB drugs, insufficient adherence of patients to treatment, and a rise in TB/HIV (human immunodeficiency viruses) co-infections during 1990–2000, drug-resistant forms of mycobacterium emerged. Belarus is among the 18 high-priority countries for TB control in the World Health Organization (WHO) European Region and among the 30 countries with a high burden of multidrug-resistant TB (MDR-TB) in the world [3].

Notably, the practice of mass chest X-ray examination screening (often called “fluorography testing or screening”) is justified in some Eastern Europe and Central Asia (EECA) countries as not only a strategy to detect pulmonary TB, but also an effective way for the early detection of LC [4]. This strategy became part of the “dispanserization” program, which was initiated in 1986 in the Soviet Union with the main focus on annual mass check-ups that are offered to the whole population. The program includes standard laboratory and additional examinations, such as chest X-ray/fluorography. In Belarus, fluorography testing is mandatory for all adults depending on age, so-called “risk group” status (which are broadly described), and whether the person belongs to an “obligatory” group. ‘Obligatory’ groups refer persons who have mandatory annual occupation-related health check-ups. Despite the existence of the dispanserization program as a public health measure, results from international randomized trials [5] fail to demonstrate the benefit of mass chest X-ray screening as a means of reducing TB or LC mortality. Program monitoring is based on the number of detected cases of the disease, including those at an early stage [6].

The burden of lung cancer (LC) is high in Belarus. The malignancy is in fourth place among all malignant neoplasms in Belarus, with 24.3 per 100,000 according to GLOBOCAN (2020). Although the causal relationship between tobacco smoking and LC was established as early as the 1950s [7], Belarus lags in implementing WHO’s “Best Buys” (recommended public health measures to reduce the prevalence of smoking). In particular, excise taxes on tobacco are insufficiently raised to the recommended minimum of 75% of the retail price, and smoke-free legislation was only adopted in 2019 [8]. Findings from the National WHO STEPwise approach to non-communicable risk factor Surveillance (STEPS) survey indicate that 48% of men and 12% of women were daily smokers [9]. The most recent Global Youth Tobacco Survey (GYTS) among 13–15 years students reported that 7% of boys and 9% of girls currently used any tobacco product in 2021, and 45% of them were not prevented from buying cigarettes in kiosks or shops because of their age [10].

The benefits of fluorography screening for LC detection in Belarus have never been assessed however, and there is a great need to investigate the advantages and disadvantages of fluorography screening in Belarus. Specifically, evaluating the yield of diagnosing TB and LC with chest X-ray/fluorography compared to an approach that is focused on the symptomatic population as part of routine passive case finding for TB and LC would be valuable from the public health perspective. The focus of this manuscript is to provide an overview of the current practices and justifications for chest X-ray screening, its terminology, and patients’ pathway through the health system. This article describes the organization of the TB service in one of the EECA countries, the documents (orders and ordinances), and the research protocol. The main study results will follow in a separate manuscript (-s) related to the LC and TB yield of chest X-ray screening. This paper will serve as a reference for the methodology of the FLUTE (FLUorography TEsting) study.

### 1.1. Fluorography: Terminology and History

Fluorography (from the Latin “fluor” to “a current, a flow” and from the Greek “grapho” to “write, represent”) is the method of X-ray inspection consisting of a photograph of the full-measured shadow image from an X-ray fluoroscopic screen or the screen of the electron-optical converter on a film of a small (i.e., 7 × 7 cm or 10 × 10 cm) format [11]. It is the photograph of the image that is produced on a fluorescent screen by X-rays [12] due to an intensification process by which some of the energy that is associated with isotope decay is converted to light by the interaction of radio-decay particles with a compound that is known as a fluor, which then exposes the X-ray film [13].

The first known use of the word fluorography appears in 1941 [12]. Other, although less used, synonyms include fluoroscopy (the original term), roentgenofluorography, cinefluorography, (X-ray) photofluorography, kymography, fluororadiography, cineradiography, videofluoroscopy, mass miniature radiography, rentgenofotographya, and photo X-ray analysis. Originally, the idea was proposed soon after the opening in 1895 of X-ray emission and followed by the development of the first fluoroscope that produced the first fluorogram (X-ray image), and in 1924 the first fluorography center for the detection of lung TB was organized in Rio de Janeiro [11].

### 1.2. Fluorography: The Scale of Implementation

In the Soviet Union, fluorography testing was introduced in the 1940s and became the main method of pulmonary TB screening in adults, thanks to the sustained state budget financing of the anti-TB program. The TB program developed a broad dispensary network, in which the main role was assigned to TB dispensaries-institutions that provided comprehensive access to TB diagnosis, treatment, rehabilitation, and follow-up of children and the adult population [14]. Fluorography increased the capacity of X-ray departments, reduced costs that were associated with inspection and beam loading on the investigated and the medical staff, and facilitated the archival of X-ray images thanks to their small format [11]. All of this contributed to the extensive use of fluorography in Belarus. Moreover, out of all medicinal radiation exposure that approximately doubled the natural background, 17.7% of the average effective equivalent doses in 1984 were attributed to fluorography [15].

Although fluorography originated as a screening tool for TB detection, due to decreasing incidence of TB [16] and increasing LC incidence and mortality within the European region as a whole and Belarus specifically [17], a population-based chest X-ray screening was continued in Belarus as LC can also be diagnosed while fluorography performed for other reasons.

There is a great need to investigate the advantages and disadvantages of fluorography screening for public health in Belarus and elsewhere. Notably, the benefits and harms of fluorography screening for LC detection have never been assessed in Belarus. Available information from the Belarus National Cancer Registry suggests a null impact on the effectiveness of fluorography for the early detection of LC [4]. According to estimates from the World Bank, the fluorography annual program costs amount to 11 million USD (out of 61 million USD annual TB budget) in Belarus and detects an estimated 12% of all TB diagnoses, while the vast majority of new active TB cases are identified through passive case findings [18].

The FLUTE study aims to determine the yield of the fluorography-based lung screening program in the detection of true LC and/or pulmonary TB cases in asymptomatic patients at the time of testing and the fraction of positive subjects without disease that are referred for further evaluation, among the subjects attending fluorography services in two rural and two urban districts in Belarus between 1 January 2015 and 31 December 2017 with positive screening results for presumed of TB or LC. The data also provide the rationale to implement improved policies and practices regarding the role of fluorography screening in the early detection of LC and TB in Belarus, other EECA countries, and elsewhere.

### 1.3. New Technology Determinants: Digitalization of Old Methods

In post-Soviet times, Belarus moved within the last decades from the analog fluorography method to the digital systems [19,20]. Currently, fluorography, as it used to be, is not performed in Belarus: all analog fluorographs have been replaced with digital X-ray equipment. For chest X-ray examinations, only plain chest radiography is performed. In the official documents regulating the procedure of the prophylactic chest X-ray examinations are called both: either chest X-ray [21,22,23] or fluorography [22] (which technically no longer exists). The Belarus Ministry of Health orders continue to reference the term ‘fluorography’ in regulating the conducting of chest X-ray examination [21,22,23]. Although technically a “digital chest X-ray examination” is performed instead of fluorography.

### 1.4. Lung Cancer and Tobacco Smoking in Belarus

LC is the most common cancer worldwide. In Belarus, LC is the fourth most common cancer in terms of morbidity but the most common in terms of mortality (GLOBOCAN 2020). According to estimates from the International Agency for Research on Cancer (IARC/WHO) (GLOBOCAN 2020), the age-specified (ASR) incidence and mortality rates are 24.3 per 100,000 and 20.1 per 100,000 people, respectively. The incidence and mortality are ten-fold more common in men than in women (53.0/100,000 and 45.1/100,000 vs. 4.9/100,000 and 3.7/100,000, respectively) [24].

### 1.5. Pulmonary (Lung) Tuberculosis: History of Tsarist Russia and the Soviet Union

At the beginning of the 20th century in tsarist Russia, and after 1922 in the former USSR until the 1930s, the calculation of the TB incidence was based on the number of people with TB symptoms seeking medical help. TB incidence amounted to 3892 per 100,000 population in 1908 [2]. In the second half of the 1930s, preventive examinations of the population began, however, data on the incidence were not published. Beginning in the late 1930s, diagnosis of respiratory TB through chest X-ray examination became widespread [25]. In 1948, by the Decree of the Council of Ministers of the USSR, the use of chest X-ray examination to detect TB acquired a mass character [25]. The decree initiated mass screening examinations, which made it possible to improve TB detection at that time, and together with other factors (improving living and working conditions after the World War II, the emergence and availability of anti-TB drugs, and other factors) contributed to a reduction of the number of registered cases in the USSR from 290/100,000 in 1950 to 34/100,000 in 1991 [1] and to reduce the mortality rate from 80/100,000 in 1941 to 34/100,000 in 1991 [2].

### 1.6. Pulmonary (Lung) Tuberculosis in Belarus, Currently Situation

After the fall of the Soviet Union in 1991 and the deterioration of the living standards of the population, increasing unemployment, and the emergence of HIV infection, and the emergence of multi-drug resistant TB (MDR-TB) in EECA countries, the level of TB incidence and mortality started gradually increasing. This happened also in Belarus. TB notifications rose from 24 cases per 100 000 in 1990 to a peak of 51 new cases per 100,000 in 2004 according to cases that were registered in the National registration system [26]. Although in some countries of the EECA, increasing TB incidence is frequently explained by a reduction of fluorography coverage in this period, National data in Belarus show consistently high levels of fluorography uptakes ranging from 72% to 87% of the general population annually between 1995 and 2017, the equivalent of annual performing 4–6 million chest X-rays/fluorography as a mass screening test [27].

In recent decades, WHO acknowledged the big success of the National tuberculosis program in the reduction of the TB incidence and mortality which, in accordance with WHO estimates (that are higher than national notification rates), dropped from 67/100,000 in 2008 to 26/100,000 in 2020, whereas the estimated TB mortality rate dropped from 11/100,000 in 2008 to 2.8/100,000 in 2020, respectively [28,29]. According to the National TB Programme, the notification rates have decreased from 48.6 per 100,000 in 2008 to 18.7 per 100,000 in 2019 [30,31]. This remarkable progress has been achieved due to the implementation of new approaches by the National tuberculosis program with the international assistance of WHO and the Global Fund while implementing the National Strategic Plan to Prevent and Control MDR-TB 2016–2020. The most significant of these approaches include: scaling up laboratory diagnostics for TB and MDR-TB using rapid molecular tests; regular updating of clinical protocols for the treatment of TB and X/MDR-TB, facilitating access to new anti-TB drugs for patients with M/XDR-TB and improving treatment results and adherence to treatment for patients with TB and MDR-TB using incentives; implementing infection control measures at the TB facilities; and expanding ambulatory model of treatment for patients with TB and MDR-TB. Acknowledging the progress that is achieved, WHO consistently recommends to Belarus reduction of mass fluorography screening in favor of the focus groups with well-documented cost-effective targeting as per the most recent WHO recommendations [32].

Despite the observed decline in TB incidence and mortality rates during the last decade, WHO still estimates 2500 TB infections annually in Belarus for 2020, including new cases, relapses, and co-infections with HIV [30]. MDR-TB is estimated to be present in 38% of newly diagnosed TB patients and 67% of previously treated TB patients [30], respectively—the highest proportions of MDR-TB that are documented in the world [32].

Patients who are considered to be “suspicious for pulmonary TB (i.e., clinical symptoms, contact with a patient diagnosed with TB, specific radiological features, etc.)” are investigated further with sputum-smear microscopy (with Ziehl–Neelsen staining) and bacteriological including drug susceptibility testing (on solid Lowenstein–Jensen or liquid medium). In addition, a rapid molecular test applies to clinical specimens, to confirm *M. tuberculosis* and exclude antibacterial resistance [21].

### 1.7. The Current Organization of Fluorography Services in Belarus

The fluorography services are organized according to the structure of primary healthcare, with X-ray rooms in each polyclinic and mobile fluorography services. Also, radiologists from the extended country’s network of TB dispensaries (Figure 1) re-evaluate fluorography for all findings that are presumptive for TB or LC (e.g., by a double reading of the images that are suspicious for lung pathology).

### 1.8. Legal Framework and Practices for Fluorography Examination in Belarus

Currently, three orders of Belarus Ministry of Health (MoH) regulate who has to be referred for fluorography testing in the country:Order (Prikaz) of the MoH 622 of 23 May 2012 “On approval of the clinical guidelines on the organization of anti-TB activities at the out-patient facilities of health care” [21].

According to Order 622, “once a year” fluorography should be indicated to the “population at special risk” of developing TB including:-Having the “social” risk of obtaining TB: homeless, migrants, released from prisons, persons living in elderly houses, alcohol addicted and drug users, and army recruits.-Having the “medical” risk of obtaining TB: HIV-positive, people with drug-related (narcological) and psychiatric disorders, diabetes mellitus, chronic gastrointestinal diseases, silicosis, chronic obstructive pulmonary disease, pleuritis, major post-TB lung residuals, cytostatic or radiological treatment, and cachexia, the ones having a period after delivery or exposed to Chernobyl radiations;-Being in contact with TB-positive contacts: e.g., in contact with infectious TB cases (at home or professionally), at farms with endemic *Mycobacterium bovis*, with prisoners or former prisoners, and two years after detention;-Former TB patients are to be checked every six months for two years after the completion of TB treatment.

2.Ordinance (*Postanovlenie*) of the Ministry of Health N74 of 29 July 2019 “On approval of the order of obligatory medical check-ups” [22].

According to Ordinance 74, “once a year” fluorography should be indicated to the “population with a potential risk of TB transmission to the community” and includes:-workers in medical facilities and elderly houses, in pharmacies and pharmaceutical industries, in educational institutions and libraries, in delivery houses, in food factories, toy factories, in the municipal sector dealing with customers (shop assistants, hairdressers), in milk/cow farms, in water supply, in hotels and hostels, in transport (taxi drivers, train stewards, etc.);-all students from the age of 17 years;

“preventive employment-related” chest X-ray examination (mandatory contingent) should be indicated.

3.Ordinance (*Postanovlenie*) of the Ministry of Health 96 of 12 August 2016 “On approval of the instruction on the order of implementing dispanserization of the population of the Republic of Belarus” [23].

According to Ordinance 96, fluorography should be indicated to:-all of the population that are aged 18–39 years every 3 years;-all of the population that are aged 40–100 years every 2 years.

Notably, digital X-ray fluorography is also widely used by primary care doctors for the (early) detection of lung pathology in symptomatic patients instead of conventional full-range X-ray.

### 1.9. Cancer Registration

According to the 2016 International Atomic Energy Agency (IAEA), the integrated mission of the Programme of Action for Cancer Therapy (imPACT) Mission Report [4], Belarus Cancer Registry (BCR) was established in 1953 and has maintained an electronic database since 1972. The BCR includes a National Cancer Registry that is located at the State Institution “N. N. Alexandrov National Cancer Centre of Belarus” (Belarussian NCC) in Minsk Region and 12 Regional Cancer Registries. Regional Cancer Registries process data that are received on all cancer patients from the respective Cancer Centres, prepare reports for the Belarus Ministry of Health, and to the management of the hospitals. The BCR covers the entire population of Belarus (2014 estimate, 9.5 million, National Statistical Institute, http://www.belstat.gov.by, (accessed on 20 July 2019)). The legislative framework concerning data collection and reporting consists of Orders by the MoH that are updated periodically (last time in 2012).

The BCR receives data on incidence from all hospitals, laboratories, and primary healthcare centers. All physicians who have diagnosed a cancer patient are obliged to send a notification to the Regional Cancer Centre, as per the residency of the patient, no later than 10 days after confirmation of the diagnosis. The peripheral LC should be confirmed by microscopy of smears or clinical material that is obtained by an intraoperative or transthoracic puncture. If the result is considered to be ineffective then atypical lung tumor resection should be performed. In case of the central LC, a tumor biopsy under bronchoscopy, followed by cytological and histological examination of the obtained material should be performed [33]. Notifications include detailed information on the patient, their diagnosis, the treatment(s) they received, and any follow-up. The main means of case finding are passive. Regional Cancer Registries receive notifications mostly on paper (through medical records, hospital discharge records, and laboratory—including pathology—results), for all newly diagnosed cancer patients from their region and also for all treatments and results from follow-up visits (including recurrences or metastases). According to the Ministry of Health Order, patients who receive cancer or carcinoma in situ (C00-C96, D00-09, ICD-10) diagnosis must be notified. The patient information is extracted from the notifications above, coded, and recorded in the cancer registration system by cancer registrars at the Regional Cancer Registries. Once a month, data that are collected by regional centers are sent to the National Cancer Registry, where they are combined to form the national database. The BCR collects all data elements that are recommended by the European cancer registries and also additional items regarding cancer disease stage, receptors, markers, treatment, and follow-up. ICD-10 and ICD-O-213 codes are applied to describe for coding topography and morphology of the tumors [4].

For mortality, the BCR obtains information (once a month) on patients who have died of cancer, and the vital status of the registered patients from Regional Cancer Centres and Regional Statistical Bureaus. The registrars visit the statistical bureaus and usually manually (imPACT 2016 Report) extract information on the cause of death and date of death from the death certificates. The National Statistical Bureau collects the number of cancer deaths by site, sex, age group, and region annually. Thus, the Ministry of Health receives information about cancer deaths from the BCR, Ministry of Statistics, and State statistical reports [4].

The BCR uses in-house developed software, to manage its database; regional cancer registries use the same platform. The BCR software includes integrated checks for consistency of data, and semi-automated linkages of records from different sources that are based upon a personal ID number, name of the patient, and the patient’s date of birth. Records that are not identical are reviewed manually and additional information is requested from the primary source, if necessary. The BCR follows IARC and European Network of Cancer Registries (ENCR) rules for reporting multiple primaries, date of diagnosis, and basis of a cancer diagnosis. The BCR uses the “tumor node metastases” (TNM) classification for the recording stage [4].

## 2. Materials and Methods

The FLUTE study aims to assess the yield of fluorography to detect true LC and pulmonary TB cases in the context of the existing population-based screening program for these diseases and to determine the fraction of positive cases without either disease referred for further evaluation. The primary objectives of the study are: to determine the detection rates of confirmed LC/TB among all subjects who received a fluorography exam; to determine the detection rates of confirmed LC/TB among asymptomatic subjects that are screened by fluorography. Secondary objectives include: to determine the false-positive rates among the screened subjects in the population; to investigate if the detection rates vary according to risk factors and groups as defined by the Ministry of Health and to compare the detection rates and stage of disease (for LC) between asymptomatic subjects that are participating in screening and subjects that are undergoing fluorography because of symptoms.

### 2.1. Study Design

This is a retrospective screening study in which subjects attending fluorography services in the cities of Minsk and Homiel and the districts Homiel and Rechytsa (a part of the Homiel region) between 1 January 2015 and 31 December 2017 with positive screening results for suspicious TB or LC will be included. From clinical records, the main reason for consulting the screening service will be obtained. The subjects will be classified as asymptomatic or symptomatic. Confirmation of the condition (TB or LC) will be retrieved from the medical records (for TB) or/and from the National Cancer and TB Registries. The proportion of subjects that are detected with TB and/or LC will be estimated according to the symptomatic status. The design of the study is summarized in Figure 2.

### 2.2. Settings

The study will be implemented in close cooperation with the researchers from the International Agency for Research on Cancer (IARC, Lyon, France).

In Belarus, the study will be conducted in the public health facilities of the capital city of Minsk (covering half of the capital city’s population of about 2 million inhabitants), the city of Homiel (approximately 500,000 people), and rural areas of the Homiel (approximately 68,000 people) and Rechytsa (approximately 66,000 people) districts (the latter is a part of the bigger Homiel region). The study settings in Belarus include:-Minsk city and Minsk Region:
○City TB dispensary No. 2, (where X-ray images from half of the capital city of Minsk are sent to be read by radiologists);○20 Minsk city polyclinic;○State Institution “N. N. Alexandrov National Cancer Centre of Belarus”;-Homiel city and Homiel Region:
○Institution “Homiel Regional Tuberculosis Clinical Hospital”;○Rechytsa Central District Hospital.

### 2.3. Participants

No patient recruitment is required. The information will be extracted from digital fluorography (PULMOSCAN X-ray system, ADANI Ltd., Minsk, Belarus) results that are available at TB dispensaries, medical charts at polyclinics, and from the records at the National Cancer (for oncological patients) and National TB (for TB patients) Registries.

The study population includes people with fluorography that is suspicious for LC and TB, who were referred to the two study’s screening centers in the cities and regions of Minsk (City TB dispensary No. 2), and Homiel (Institution “Homiel Regional TB Clinical Hospital”) between 2015 and 2017. The medical charts of those subjects that are suspected of LC or TB within the scope of the fluorography screening program will be checked retrospectively, to identify: (i) the confirmation of the LC or TB diagnosis; (ii) the main reason for their referral for fluorography testing (e.g., were the subjects symptomatic or asymptomatic at the referral). Reasons for referral include: (i) doctor’s indication due to clinical symptoms; (ii) self-referral by the patient; (iii) for employment-related fluorography (mandated by ordinance N74 from 29 July 2019 [22]); and (iv) by a doctor’s indication as belonging to TB risk groups.

### 2.4. Eligibility and Exclusion Criteria

Subject records that are eligible for inclusion are those with:-fluorography-positive findings that are suspicious for lung LC visualized by screening fluorography X-ray, or-fluorography-positive findings that are suspicious for TB visualized by screening fluorography X-ray and were referred for additional examinations to be performed in oncological or TB facilities, respectively

Records that were excluded from the study include:-subjects who are fluorography-negative (no findings due to fluorography testing),-subjects with fluorography-positive findings that are not suspicious for LC and TB (other pathology),-subjects who were referred for additional examinations but did not go, or-subjects with the disease that was diagnosed before the fluorography test or its recurrence.

### 2.5. Duration

IARC scientists offered their help for the study design, surveys for data collection as well as the platform for data storage and its future analysis (discussed with IARC, WHO). The project was performed in 2018, the data were collected for 2015–2016, and data analysis was made at IARC.

### 2.6. Data Sources

Data from Belarus TB dispensaries (fluorography results, data of the study) and policlinics (medical charts) will be extracted by trained medical doctors onto a standardized data collection form that was designed for the FLUTE study using REDCap (research electronic data capture) software. Similarly, an oncologist that was affiliated with the National Cancer Centre abstracted LC data from the National Cancer Registry and entered the data into the study’s standardized REDCap data collection form.

Specifically, data collectors obtained the lists of subjects who were detected suspicious for LC or/and TB between 2015 and 2017 in the two screening centers in Minsk and Homiel that were included in the study. Then, the doctors search each subject (first/patronymic/last names, date of birth, and gender) on medical records and load the retrieved clinical information to a REDCap platform, on a project that is specially designed for the FLUTE study. Additional information that is extracted includes: the date of fluorography (the age would be calculated automatically based on the date of birth), occupation, smoking status, Belarus fluorography screening population group (i.e., risk group category if applicable), radiologist’s conclusion (fluorography result), diagnosis confirmation (from the medical charts and, in case of LC, such data should be cross-checked with the records at the National Cancer Registry and information on staging and grading to be added by the fifth separate data collector (who is an oncologist from the National Cancer Centre and Primary Investigator), histology results and data, symptoms (e.g., coughing, sweating, chest pain, weight loss, hemoptysis, chest pain, shortness of breath), treatment results (if applicable), date of death (if applicable), and a field for additional comments for data collectors could be provided.

Only anonymized data will be entered into the REDCap platform. The platform is displayed in Russian and all data entry is done in this language. The database relies on codes that have a corresponding English label. The data collector comments will be translated into English before the analysis, if necessary.

## 3. Study Management

The study and database designs were developed at the Prevention and Implementation Group (PRI) of the Section of Early Detection and Prevention (EDP) at the IARC. The IARC is a specialized agency of the World Health Organization focusing on different aspects of cancer research.

Fluorography data on suspected LC and/or lung TB cases are available at TB dispensaries in Minsk (one out of two) and Homiel (single-center), respectively. Both institutions, equipped with fluorography machines, have routinely been conducting fluorography-based early diagnosis and screening in Belarus.

## 4. Statistical Analysis

### 4.1. Sample Size Calculation

Between 2014 and 2016, 1,440,534 chest X-ray images were read by radiologists in TB dispensary No.2 in Minsk from the population of approximately 996,343 people living in Minsk (half of the city’s population of 1,992,685 people [34], are served by the TB dispensary No.2). This means that an average of 480,178 images were read annually. By including data for 2017, the total sample represents approximately 1,920,712 images for the years 2014 and 2017. In addition to Minsk, data from three more locations with a total population of 670,918 (including Homiel city (535,693 inhabitants [34]), Homiel district (68,306 [34]), and Rechytsa district (66,919 [34]), respectively) were included in the study sample. Based on the data that were available from Minsk, approximately 970,028 images from this population could be read between 2014 and 2017, making the total number of images that were read within this period in the study locations 2,410,562. On average, among the read fluorography images, 0.016% and 0.04% were considered screen-positive for suspected TB and LC cases, respectively. Therefore, among the 2,410,562, we expect 386 and 964 suspected for TB and LC, screen-positive, images.

Taking into account that the majority of lung nodules that were detected by chest computer tomography (no data for X-ray fluorography are available), typically over 90%, will be benign, even in a targeted, high-risk population [34], LC will be confirmed in less than 10% of suspicious for LC cases. The positive predictive value for detecting culture-positive TB on digital chest X-ray films is 11% [34]. Therefore, among the suspected (screen-positive) TB and LC images, we expect 43 (11% out of 386) and 96 (10% out of 964) confirmed TB and LC cases, respectively.

Therefore, the expected the fluorography detection rate among the 2,410,562 images will be 0.002% for TB and 0.004% for LC for the period 2014–2017, respectively. With a sample size of 2,410,562 images and a Type I error of 0.05, the precision for the interval estimation of the detection rate of TB and LC is 0.0006% and 0.0008%, respectively.

### 4.2. Statistical Analysis

We will perform descriptive analyses to describe the study population and conduct chi-square tests to determine the statistical differences between the proportion of TB and LC detection according to the symptomatic status (symptomatic vs. asymptomatic). Using unconditional logistic regression, we will determine if the proportion of the disease condition (TB or LC) in symptomatic subjects is higher or lower than among asymptomatic subjects. Additionally, using unconditional logistic regressions, we will assess if the characteristics of the screened population (i.e., special risk) modify the likelihood/chance of being diagnosed with the disease condition (TB or LC). All of the analyses will be done for TB and LC separately. A *p*-value of 0.05 will be considered for significant differences. Stata will be used for all the statistical procedures.

### 4.3. Data Management

All data will be securely stored and made available for audit and collaborative research according to the standard procedures of the collaborative centers and IARC Information Security Policy.

Anonymized data containing these results as well as information on clinical symptoms from medical charts have been stored on the REDCap platform for further statistical analysis. The storage and archival of FLUTE study data adhere to applicable Belarus regulations for IARC and the decision of the Belarus Ethical Committee. The statistical analysis is to be performed by IARC statisticians ensuring the ethics process is respected.

### 4.4. Consent and Ethics Considerations

The study collects data on oncological patients that are prospectively scheduled for routine diagnostic chest X-ray examinations at the Minsk and Homiel TB dispensaries as part of their routine national medical check-ups. Thus, no written informed consent was provided or requested.

In Belarus, ethical approval was granted on 2 November 2018 (No. 149), by the Ethical committee of the State Institution “N. N. Alexandrov National Cancer Centre of Belarus”, which is the national scientific institution for oncology. At IARC, ethical approval was granted on 9 October 2019 (IEC Project No. 19-30)

### 4.5. Preliminary Results: Lung Cancer Detection, 2015–2017

To illustrate the utility of FLUTE study data for program evaluation, we present preliminary results on LC detection during the years 2015–2017 from the pilot regions of Minsk city, Homiel city, Homiel district, and Rechitsa district. We prioritized reporting preliminary LC results first, before pulmonary TB results, because of the high incident mortality rates: 24.3 per 100,000 and 20.1 per 100,000 people (ASR), respectively (GLOBOCAN 2020), associated with LC. Moreover, compared with LC, pulmonary TB is a treatable disease in Belarus (or in CEEA). Notably, these preliminary data must be interpreted with caution as we have not evaluated the role of potential confounding variables, nor do we report the background demographic characteristics of the study population.

During 2015–2017, 3043 LC patients were registered in BCR at four FLUTE study locations. Of these patients, 760 (25%) LCs were diagnosed through fluorography screening. Among these 760 patients, 486 (63.9%) presented with symptoms at the time of their fluorography screening exam and 274 patients that were detected through fluorography screening were asymptomatic at the time of their diagnosis. Among the 274 asymptomatic LC cases, one case was not histologically confirmed. Of the remaining 273 histologically confirmed LC cases, 42.8% (117 out of 273) patients were diagnosed in the early (I or II) stages of the disease. Therefore, 3.8% (117 patients) of all the registered LC patients at the FLUTE study project locations were detected with the asymptomatic disease through fluorography screening exams (Figure 3).

## 5. Discussion

The FLUTE study represents a collaborative endeavor that is coordinated by WHO and the International Agency for Research on Cancer among Belarus institutions that are responsible for LC prevention, screening, and treatment in Belarus; the national Tuberculosis Control programme and registry; and the Ministry of Health in Belarus to evaluate the effectiveness of population-based fluorography. Notably, the study presents the first opportunity to assess this approach for the diagnosis of LC using a population-based chest X-ray examination. The FLUTE study will analyze results from a sampling frame that is drawn from more than 2.4 million images that were recorded within national cancer and TB registries during a three-year period.

The purpose of this paper is to provide the background details of the FLUTE study methodology and to serve as the reference for the future manuscript (-s) that present the study results. For illustrative purposes, we present preliminary data on the overall LC detection that was collected through the study. Despite the preliminary nature of these data, they highlight how FLUTE data provides practical information for program evaluation and can be useful to guide policy-making with regard to LC screening.

The current practice of mass fluorography-based screening in Belarus reflects a continuation of earlier USSR legislation mandates; while such practices have become outdated and obsolete, they often persist without robust evaluation of their real effectiveness, as new and more accurate tools have become available. Thus, FLUTE study findings have the potential to inform other countries and programs that rely on population-based, fluorography-based screening approaches for LC and TB detection. Additionally, earlier mandated practices may no longer align with the current epidemiological context. As the epidemiology of LC and TB evolves in Belarus and the region, effective and evidence-based screening and detection tools should be used as a public health principle.

Further, the FLUTE study aims to provide clarity on two important public health concepts as they relate to LC and TB: screening and early detection. Historically, fluorography was established as a screening tool for TB infection; however, when the incidence of LC became a widespread disease, the medical community began using fluorography as a means to detect LC. Frequently, mass-health check-ups (or dispanserization) and early detection are used interchangeably; however, there are important and distinct differences between these concepts. Screening refers to the identification of a disease in the general population without symptoms; whereas, the purpose of early detection is to identify disease in populations that are at high risk. Current documents often misuse terminology that is related to fluorography and thus introduce confusion as to the interpretation of image results.

We envision there are several strengths of the FLUTE study that will support the interpretation of the study results. Firstly, the study uses population-based data from the most prominent urban area in Belarus as well as another area in the country that includes both urban and rural settings, to minimize inequities due to geographic differences. Moreover, images that were obtained from the dispensary in Minsk represent more than 50% of the city’s population. We abstract data for multiple years to ensure a robust sample of symptomatic and asymptomatic detected cases of LC and TB to inform public health deliberations. The study’s time period sufficiently captures multiple screening cycles, as persons that are deemed at social risk are required to be screened every two years. Standardized data abstraction instruments will be used with rigorous data management oversight to assure high-quality data. The study relies on complete, validated registry data that adheres to global reporting standards. Execution of the FLUTE study draws upon research leadership by IARC with its longstanding record of conducting large multi-center intra-disciplinary studies in the fields of cancer prevention, screening, and early detection, and has outstanding expertise in cancer epidemiology. Further, the study builds upon a collaborative team of scientists at IARC, WHO Belarus, and the leading Belarus Technical Institutes.

We acknowledge there are several limitations of the FLUTE study that warrant mention. Firstly, due to its retrospective design, several biases may affect the interpretation of the FLUTE study findings. For example, FLUTE data are abstracted from registry databases, which may have missing records and/or incomplete data. While the National Cancer Registry adheres to Global (IARC) and European recommended reporting standards, it is not possible to verify the accuracy of data recorded in the registry, such as diagnostic status. Additionally, the study population includes both asymptomatic and symptomatic participants, thereby introducing confounding factors concerning interpreting and evaluating the role of fluorography for screening purposes. Given the study’s sole focus on patients that are diagnosed with either LC or TB, we will not be able to determine whether patients that are diagnosed with other conditions, such as pneumonia, are asymptomatic of LC or TB. As with all retrospective studies, it will not be possible to evaluate the extent to which unmeasured confounding influences, due to another factor (such as pneumonia or another co-morbidity), influences any results that are gleaned from this study. Lastly, we recognize that evidence from the FLUTE study may not be entirely generalizable to the entire population of Belarus. However, FLUTE data represents screening data that are reported from the largest urban area in Belarus (Minsk) and another region (Homiel) that contains both urbanized and rural areas within its catchment.

Nevertheless, the FLUTE study will be uniquely positioned to provide evidence on whether the current mass application of fluorography screening identifies incidental LC or TB cases as (or more) effectively compared with the targeted screening approach among high-risk populations. Reports using all study data will be published in peer-reviewed scientific journals, as open access. Opportunities to share study results, both in English and Russian, with key stakeholders through meetings, webinars, and scientific forums will be organized. Additionally, technical support and access to study data will be offered to local investigators who express interest in leading further analyses of the FLUTE study.

## 6. Conclusions

The FLUTE study will offer evidence regarding whether the use of X-ray fluorography for mass TB and LC detection adds value or whether different, targeted screening approaches are more appropriate from both clinical and public health perspectives. The project responds to programmatic and policy questions facing national TB control and cancer control programs. The FLUTE study contributes to cancer control, as it evaluates a national program on early detection of LC and provides scientific evidence on its validity in a middle-income country.

## Figures and Tables

**Figure 1 ijerph-19-08706-f001:**
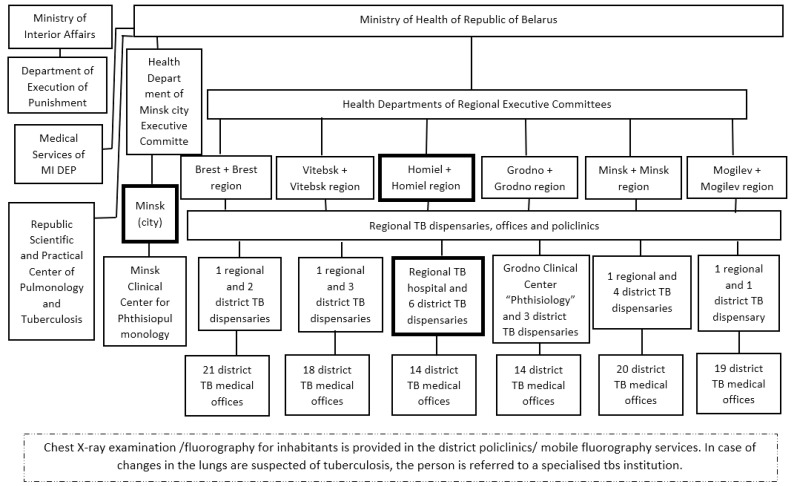
The structure of tuberculosis service in the Republic of Belarus. Footnote: The vertical structure of TB services (TB dispensaries) in each country region is divided by districts. (Boxes with bold boundaries indicate the districts and city (Minsk) where data were collected for the project).

**Figure 2 ijerph-19-08706-f002:**
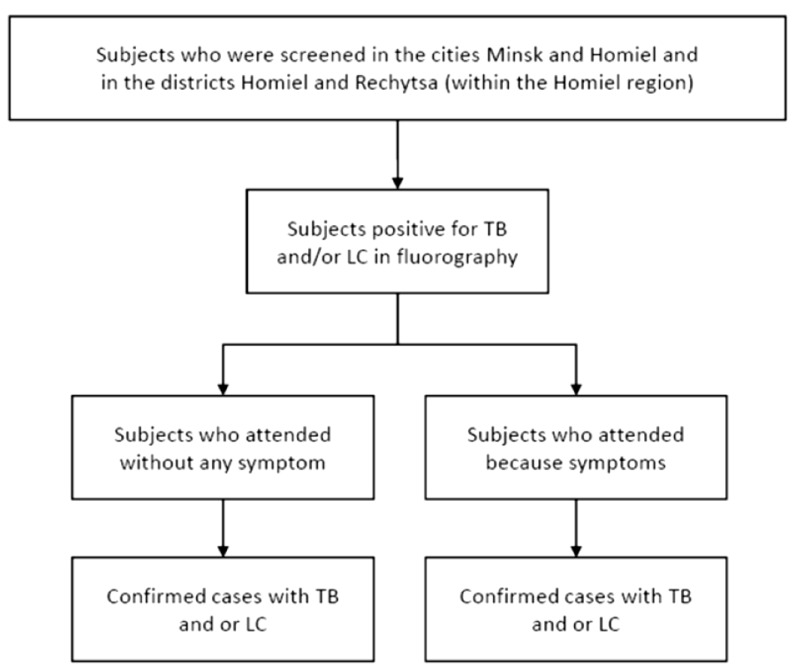
The design of the FLUTE study.

**Figure 3 ijerph-19-08706-f003:**
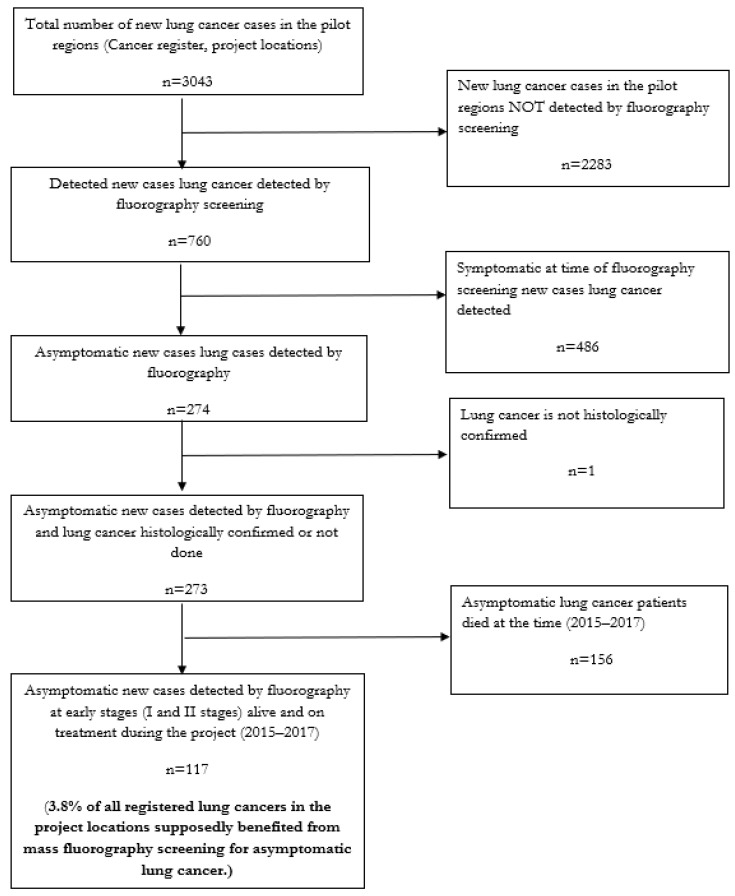
The proportion of asymptomatic cases of lung cancer that was detected at early stages (I and II stages) by fluorography preventive mass screening out of all the registered lung cancer cases in the FLUTE study locations of Minsk city, Homiel city, Homiel district, and Rechitsa district (Belarus) 2015–2017.

## Data Availability

Not applicable.

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
