# Peer review of "Rationale and Purpose: The FLUTE Study to Evaluate Fluorography Mass Screening for Tuberculosis and Other Diseases, as Conducted in Eastern Europe and Central Asia Countries"

_ijerph, 2022, doi:10.3390/ijerph19148706_

Round 1
Reviewer 1 Report
Thanks for recommending me as a reviewer. The paper is a retrospective review of medical records to assess the yield of fluorography to detect true cases of LC and/or TB in asymptomatic patients in two rural and two urban districts in Belarus for 2015-2017 with positive screening results for presumed of TB or LC. The study provided the rationale to implement the improved policy and practices regarding the fluorography role in the early detection of LC and TB in Belarus and elsewhere. If authors complete minor revisions, the quality of the study will be further improved.
1. The introduction section is well written. If authors combine the last paragraph of the introduction section with the previous one, the quality of the study will be further improved.
2. Please correct the typographical errors. ex. [[19], [20]]
3. If authors combine the last sentence of the conclusion section with the previous paragraph, the quality of the study will be further improved.
Author Response
The 1st reviewer
Comments and Suggestions for Authors
Thanks for recommending me as a reviewer. The paper is a retrospective review of medical records to assess the yield of fluorography to detect true cases of LC and/or TB in asymptomatic patients in two rural and two urban districts in Belarus for 2015-2017 with positive screening results for presumed of TB or LC. The study provided the rationale to implement the improved policy and practices regarding the fluorography role in the early detection of LC and TB in Belarus and elsewhere. If authors complete minor revisions, the quality of the study will be further improved.
RE: Thank you for your positive review, which is much appreciated. We have accepted all your suggestions and incorporated them into a revised manuscript.
- The introduction section is well written. If authors combine the last paragraph of the introduction section with the previous one, the quality of the study will be further improved.
RE: We agree, thank you. The paragraphs were combined (lines 49-57).
- Please correct the typographical errors. ex. [[19], [20]]
RE: We have corrected the grammatical/typing errors across the manuscript, thank you for noticing it.
- If authors combine the last sentence of the conclusion section with the previous paragraph, the quality of the study will be further improved.
RE: We agree, thank you. The last sentence of the conclusion section was combined with the previous paragraph (lines 538-540).
Reviewer 2 Report
In this retrospective study authors review medical records to assess the yield of fluorography to detect true cases of LC and/or TB in asymptomatic patients in two rural and two urban districts in Belarus for 2015-2017. The study provides rationale to implement the improved policy and practices regarding the fluorography role in the early detection of LC and TB. Manuscript is well written highlighting all the limitations of the study. My comments are below.
1) The preliminary lung cancer data is providing the rationale to implement the improved policy and practices regarding the fluorography; however, it lacks potential confounding variables and background demographic characteristics which is a concern and has been noted by the authors.
2) What laboratory tests were performed to confirm TB or LC in asymptomatic subjects should be mentioned.
3) Full form of some of the abbreviation such as FLUTE is missing. Recommend authors to recheck the manuscript for every abbreviation used in the text.
4) Manuscript should be rechecked for grammatical/typing errors.
Author Response
The 2nd reviewer
Comments and Suggestions for Authors
In this retrospective study authors review medical records to assess the yield of fluorography to detect true cases of LC and/or TB in asymptomatic patients in two rural and two urban districts in Belarus for 2015-2017. The study provides rationale to implement the improved policy and practices regarding the fluorography role in the early detection of LC and TB. Manuscript is well written highlighting all the limitations of the study. My comments are below.
RE: Thank you for the positive review which is much appreciated. We hope that we could fully address your suggestions and comments, which have definitely improved the manuscript. Thank you.
- The preliminary lung cancer data is providing the rationale to implement the improved policy and practices regarding the fluorography; however, it lacks potential confounding variables and background demographic characteristics which is a concern and has been noted by the authors.
RE: We greatly appreciate the thoughtful perspectives and suggestions from the reviewer, which will take to heart and incorporate into the manuscript with results from the study. The current manuscript describes the rationale and purpose of the protentional studies which could be proposed to be conducted to evaluate fluorography mass screening programmes for pulmonary tuberculosis and other lung diseases ongoing in several Eastern Europe and Central Asia countries, and beyond. Thank you for the opportunity to improve our next manuscript, where such analyses would take place for Belarus, a country where the healthcare system steel to face to multiple changes.
2) What laboratory tests were performed to confirm TB or LC in asymptomatic subjects should be mentioned.
Thank you for your insight. We have added some additional information, please see the following text
- Lines 172-177, within the chapter 1.5 Pulmonary (lung) tuberculosis in Belarus, currently situation: “Patients who are considered to be “suspicious for pulmonary TB (i.e., clinical symptoms, contact with a patient diagnosed with TB, specific radiological features, etc.)” are investigated further with sputum-smear microscopy (with Ziehl-Neelsen staining) and bacteriological including drug susceptibility testing (on solid Lowenstein-Jensen or liquid medium). In addition, a rapid molecular test applies to clinical specimens, to confirm M. tuberculosis and exclude antibacterial resistance (According to the national Order of the MoH 622 of 2012).”
- Lines 242-246, within the chapter 1.8 Cancer registration: “The peripheral lung cancer should be confirmed by microscopy of smears or clinical material obtained by an intraoperative or transthoracic puncture. if the result is considered to be ineffective then atypical lung tumor resection should be performed. In case of the central lung cancer, a tumor biopsy under bronchoscopy, followed by cytological and histological examination of the obtained material should be performed (according to the Ordinance (Postanovlenie) 60 of the MoH of 2018)”.
- Full form of some of the abbreviation such as FLUTE is missing. Recommend authors to recheck the manuscript for every abbreviation used in the text.
RE: Duly noted, thank you very much. We have provided full forms for the abbreviations used in the manuscript, such as: FLUTE, HIV, WHO, STEPS, REDCap.
- Manuscript should be rechecked for grammatical/typing errors.
RE: We have corrected the grammatical/typing errors across the manuscript, thank you for noticing it.

This manuscript is a resubmission of an earlier submission. The following is a list of the peer review reports and author responses from that submission.
Round 1
Reviewer 1 Report
Thanks for recommending me as a reviewer. The study provided the rationale to implement the improved policy and practices regarding the fluorography role in the early detection of lung cancer (LC) and tuberculosis (TB) in Belarus and elsewhere. If the authors complete the revision, the quality of the study will be further improved.
1. The abstract is intended for theoretical background and purpose only. If authors add methods, results, and conclusions to the abstract, it will be helpful for readers to understand.
2. line 68-71: A paragraph cannot consist of a single sentence. It is recommended that the last paragraph of the introduction section be combined with the previous paragraph.
3. The results of the study were not presented.
4. The limitations of the study should be added in the discussion section.
Reviewer 2 Report
The authors present the FLUTE study as a collaborative endeavor coordinated by WHO and the IARC among Belarus institutions responsible for lung cancer prevention, screening, and treatment in Belarus; the national Tuberculosis Control program and registry; and the Ministry of Health in Belarus to evaluate the effectiveness of population-based fluorography. In Belarus, periodic population-based chest X-rays use as a mass screening tool for the diagnosis of tuberculosis, often justified for early detection of lung cancer. However, as the authors said, no mortality benefits were demonstrated by screening with chest X-rays in international randomized trials, and I am not sure it exists. The authors want to assess this approach for the diagnosis of lung cancer using population-based chest X-rays. They analyzed the results from a sampling frame drawn from more than 2.4 million images recorded within national cancer and tuberculosis registries over three years. The study assesses the yield of fluorography to detect true cases of LC and/or TB in asymptomatic patients in two rural and two urban districts in Belarus from 2015-to 2017 with positive screening results for presumed TB or LC.
The current practice of mass fluorography-based screening in Belarus has become obsolete, and more accurate tools have become available. They want to inform populations of other countries that practices are outdated. However, the work is a simple presentation of the study, and some results are waited to evaluate the interest of the study. The authors should discuss the published data which evidence the proportion of lung cancers diagnosed/detected by radiography at early stages. Other tools are also available such as circulating tumor cells, or tumoral circulating DNA, as well as some proteins biomarkers. Indeed, X-rays chest for TB diagnosis should also be compared and discussed with some bacteriological test, maybe costless. What impact has repetitive fluorography in terms on exposure to the body? What is the incidence of radiation-induced cancer?
Lots of terms are not defined (EECA, mln…..).
Round 2
Reviewer 2 Report
Thank you for providing a revised manuscript.
Even if the authors have provided a revised version of their work, I am still not convinced of the interest of this study. I am still needed some preliminary results to find this work relevant for publication. Sorry for keeping my initial position.